# Analysis of DIA proteomics data using MSFragger-DIA and FragPipe computational platform

Fengchao Yu [1] ✉, Guo Ci Teo [1], Andy T. Kong[1,2], Klemens Fröhlich [3], Ginny Xiaohe Li[1], Vadim Demichev[4,5] & Alexey I. Nesvizhskii [1,2] ✉

Liquid chromatography (LC) coupled with data-independent acquisition (DIA) mass spectrometry (MS) has been increasingly used in quantitative proteomics studies. Here, we present a fast and sensitive approach for direct peptide identification from DIA data, MSFragger-DIA, which leverages the unmatched speed of the fragment ion indexing-based search engine MSFragger. Different from most existing methods, MSFragger-DIA conducts a database search of the DIA tandem mass (MS/MS) spectra prior to spectral feature detection and peak tracing across the LC dimension. To streamline the analysis of DIA data and enable easy reproducibility, we integrate MSFragger-DIA into the FragPipe computational platform for seamless support of peptide identification and spectral library building from DIA, data-dependent acquisition (DDA), or both data types combined. We compare MSFragger-DIA with other DIA tools, such as DIA-Umpire based workflow in FragPipe, Spectronaut, DIA-NN library-free, and MaxDIA. We demonstrate the fast, sensitive, and accurate performance of MSFragger-DIA across a variety of sample types and data acquisition schemes, including single-cell proteomics, phosphoproteomics, and large-scale tumor proteome profiling studies.

Liquid chromatography (LC) coupled to data-independent acquisition (DIA) mass spectrometry (MS) strategy has emerged as a widely used technological platform for quantitative protein profiling, especially in large-scale studies[1,2]. Compared to the data-dependent acquisition (DDA) MS strategy, in which a selected peptide ion population is isolated and subjected to tandem MS (MS/MS) fragmentation, the fragment ion information in DIA is acquired on all peptide ions within a certain window of m/z values, sequentially covering the entire relevant range (e.g., 400–1200 Da). This helps alleviate the stochastic nature of peptide identification in DDA-based strategies, and DIA has been shown to produce more complete (i.e., less missing values) peptide and protein quantification matrices across multiple samples[2]. DIA has been successfully used in a variety of proteomics applications, including post-translational modification (PTM) analysis[3–6],

protein–protein interaction[7,8], immunopeptidomics[9–11], and heritability analysis[12].

MS data used as part of a DIA analysis workflow can be categorized into primary and auxiliary data. Auxiliary MS data is used solely to build a spectral library for subsequent targeted extraction of quantification for each peptide ion in the library from the primary DIA data. Examples of such auxiliary, "library-only" MS data include data acquired from pooled samples (e.g., all individual samples profiled in the study) fractionated using offline LC and analyzed using DDA[2], fractionated in the gas phase and analyzed using narrow-window DIA (GPF-DIA)[13,14]. For low-input proteomics, including single-cell proteomics, library-only data can be acquired in DDA or DIA mode from samples prepared from higher amounts of starting material[15–17]. The primary DIA data are DIA runs acquired on individual study samples, typically without

[1]Department of Pathology, University of Michigan, Ann Arbor, MI, USA. [2]Department of Computational Medicine and Bioinformatics, University of Michigan, Ann Arbor, MI, USA. [3]Proteomics Core Facility, Biozentrum, University of Basel, Basel, Switzerland. [4]Department of Biochemistry, Charité – Universitätsmedizin Berlin, Berlin, Germany. [5]Department of Biochemistry, University of Cambridge, Cambridge, UK. ✉e-mail: yufe@umich.edu; nesvi@med.umich.edu

fractionation, although the use of fractionation has been explored[18]. The primary DIA data are used for extracting peptide ion quantification, but they may also be used for building the spectral library, either alone or in combination with auxiliary DDA or DIA data[19].

The computational analysis of DIA data has two major components: (1) creation of the target spectral library, a collection of peptide ions that are targets for the subsequent quantification step, along with their LC retention times (RT), as well as m/z values and intensities of the corresponding fragment ions; (2) extraction of quantification from the primary DIA data for all peptide ions in the target spectral library. Quantification is performed using targeted extraction tools, such as Skyline[20], OpenSWATH[21], EncyclopeDIA[13], Spectronaut[22], and DIA-NN[23], with additional intensity data normalization and peptide-to-protein rollup[24,25]. The targeted quantification step depends on the quality of the input spectral library[26]. An ideal library would be as experiment specific as possible, that is, it would be complete (containing all peptide ions that are likely to be detectable in the analyzed samples) and as specific (i.e., it would not contain unrelated peptides) as possible[27,28]. Library information, such as the RT and fragment ion intensities, should match the actual DIA data being analyzed well. Thus, building spectral libraries via the direct identification of peptides from primary DIA data[19] has emerged as a widely used computational data analysis strategy.

Direct identification of peptides from DIA data was first proposed and implemented in DIA-Umpire[19], along with the concept of building combined (hybrid) spectral libraries from primary DIA and auxiliary (e.g., DDA) data. The "spectrum-centric" identification approach of DIA-Umpire is based on feature detection in MS1 and MS/MS data, followed by grouping precursor peptide and fragment ion signals exhibiting similar LC elution profile to generate the so-called pseudo-MS/MS spectra. These pseudo-MS/MS spectra can be searched using tools developed for conventional DDA data, such as MSFragger[29], X! Tandem[30], and Comet[31], followed by peptide-spectrum match (PSM) validation with PeptideProphet[32] or Percolator[33], protein inference with ProteinProphet[34], and target-decoy based false discovery (FDR) filtering[35]. The DIA-Umpire strategy has been increasingly adopted in other tools and pipelines, including Spectronaut's directDIA. A drawback of this strategy is that the untargeted signal extraction of the precursor and fragment ion peak curves from DIA MS1 and MS/MS scans can be time-consuming. The sensitivity of the peptide identification process may also be suboptimal compared to targeted peptide detection methods. Other strategies for the direct identification of peptides from DIA data have also emerged, often taking the peptide-centric perspective[36] to the same problem, as exemplified by tools such as PECAN[37] (Walnut in EncyclopeDIA[13]). Direct search of DIA MS/MS spectra against repository-wide spectral libraries or sequence databases has also been explored[38,39]. The ability to predict proteome-wide spectral libraries using deep learning[40–46], followed by the creation of more refined, experiment-specific libraries for targeted extraction, has also been shown to be very useful[47], and formed the basis for the in silico library-based (also known as "library-free") mode of DIA-NN[23,48], EncyclopeDIA[13], and MaxDIA[49]. However, only peptides with common modifications (such as oxidation, acetylation, and phosphorylation) can typically be predicted using the current in silico spectral library software. Furthermore, workflows based on proteome-wide spectral library prediction suffer from additional limitations, such as long prediction times or requirements for additional graphics processing units (GPUs).

We develop a new approach for direct peptide identification from DIA data, leveraging the unmatched search speed of fragment ion indexing[29]. It is based on conducting a database search of DIA MS/MS spectra prior to feature detection or peak tracing, blurring the difference between the analysis of DIA and DDA MS/MS spectra. It is implemented as MSFragger-DIA, a separate module in the MSFragger search engine[29,50,51], and publicly available since MSFragger version 3.1

(released on September 30, 2020). Using MSFragger software, one can identify peptides from either DDA or DIA data, or from both data types combined, allowing the seamless generation of a hybrid spectral library for the most sensitive analysis. We compare MSFragger-DIA with other tools, such as DIA-Umpire based workflow in FragPipe, Spectronaut, DIA-NN library-free, and MaxDIA in MaxQuant. We demonstrate the fast, sensitive, precise, and accurate performance of MSFragger-DIA across a variety of sample types and acquisition schemes, including single-cell proteomics and phosphoproteomics applications. To lower the barrier of analyzing DIA data, we fully integrate MSFragger-DIA into the FragPipe computational platform (https://fragpipe.nesvilab.org/). In tandem with the spectral library building module EasyPQP[48] and the quantification tool DIA-NN, MSFragger-DIA in FragPipe enables the complete analysis of DIA data, from peptide identification to peptide and protein quantification.

## Results
### MSFragger-DIA algorithm
An overview of the algorithm and its implementation within the FragPipe computational platform is shown in Fig. 1. MSFragger-DIA starts with a direct search of MS/MS spectra against the entire protein sequence database, before any peak tracing or feature detection procedures (Fig. 1a). MSFragger-DIA starts with deisotoping[50] and extraction of the isolation window information from the spectra. Because no precursor charge information is available for DIA MS/MS spectra, it enumerates all predefined charge states to calculate the lower and upper precursor mass bounds for each spectrum. Each spectrum is then searched against all database peptides within the precursor mass range. To calibrate fragment masses, MSFragger-DIA performs two searches: a fast calibration search and a full search. The first, more restricted search (which considers peptides with precursor charge states 2 and 3 only) is used to select high-quality spectra to build a mass error profile and perform mass calibration, as described previously[51]. MSFragger-DIA then performs a full search using the calibrated data. The outcome of this step is a list of peptide candidates (by default, 128 best-scoring peptides per spectrum in wide-window DIA data; 32 in narrow-window GPF-DIA data), which is then further refined as described below.

In the second step, MSFragger-DIA traces peaks, extracts ion chromatograms, and detects features of all candidate peptides for each spectrum determined as described above. This can be done quickly using the MS1 and MS/MS spectral indexing technology that we described previously[52]. For each peptide, MSFragger-DIA refines the list of matched fragments that can contribute to the score. The apex LC retention times of the fragments and precursor features are collected, and the median retention time is calculated. Fragments with retention times that differ from the median by more than a certain value (0.1 min by default) are filtered out. If a precursor ion is detected in the MS1 data and has an apex retention time outside the allowable range, the PSM is filtered out. Note that if no precursor ion feature is detected in the MS1 spectrum, MSFragger-DIA still retains the corresponding PSM, as there are cases where fragment ion signals exist despite missing (undetectable) precursor features. MSFragger-DIA also filters out PSMs with aberrant isotope distributions[53]. After peak tracing and peptide and fragment filtering, MSFragger-DIA normalizes and rescores peptide matches using hyperscore[29,30].

Since each MS/MS spectrum is matched against candidate peptides independently, the same experimental peaks may match and contribute to the score of multiple different peptides in the candidate peptide list; a greedy method is used to prevent this from happening (Fig. 1b). Given a spectrum and a list of candidate peptides, MSFragger-DIA selects the highest-scoring peptide and removes matched fragment peaks from the spectrum. Then, it normalizes the remaining peaks and rescores the peptides on the candidate list to obtain the second highest scoring PSM. The matched peaks are again removed,

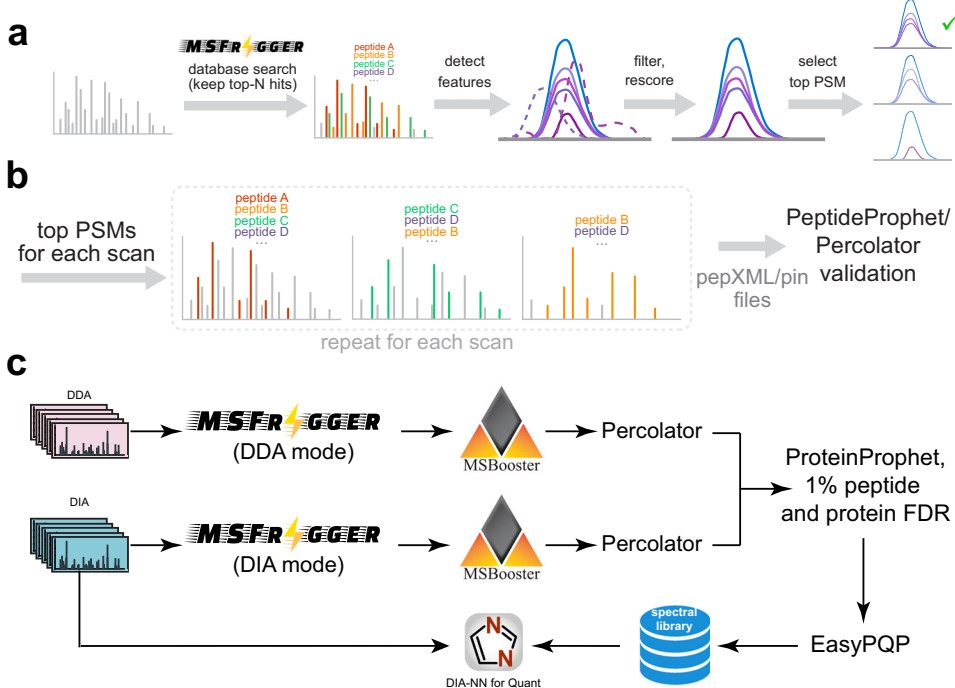

**Fig. 1 | Overview of MSFragger-DIA in FragPipe. a** DIA spectra are searched by MSFragger-DIA directly using precursor candidates determined from the isolation window. MSFragger-DIA builds MS1 and MS/MS spectral indexes, which are then used to detect extracted ion chromatogram (XIC) features for all fragment and precursor peaks in a peptide-spectrum match (PSM). Noisy fragment peaks are filtered out based on the XIC, PSMs are rescored, and only the top scoring PSM from each feature is kept. **b** Within each DIA MS/MS scan, a greedy method is used to remove matched peaks and iteratively rescore peptide candidates from the top-N list. Finally, MSFragger-DIA generates pepXML and pin files for PeptideProphet and Percolator to estimate the peptide probability in FragPipe. **c** Hybrid (combined DIA and DDA) data analysis workflow in FragPipe ("FP-MSF hybrid" in the main text). Both DDA and DIA data are used to build a combined spectral library. This spectral library is used to quantify peptides from the DIA data using DIA-NN.

peptides are rescored, and the process continues until there are fewer than four peaks left in the spectrum, or no peptide candidates remained. Finally, MSFragger-DIA generates output files compatible with PeptideProphet and Percolator for rescoring and FDR estimation.

MSFragger-DIA has been fully integrated into FragPipe, allowing one-stop DIA data analysis (Fig. 1c and Supplementary Fig. 1). In FragPipe, the output of MSFragger-DIA and MSFragger can be processed by MSBooster[54] to leverage additional scores using deep-learning prediction. The output from MSBooster is fed to Percolator for additional rescoring and posterior error probability calculation, followed by ProteinProphet[34] for protein inference, Philosopher[55] for false discovery rate (FDR) filtering, and spectral library building with EasyPQP[48]. With FragPipe, users can build the libraries from DDA (with MSFragger), DIA (with MSFragger-DIA), or all data combined (hybrid library). The resulting library is then used to extract peptide ion quantification from the primary DIA data using DIA-NN, which is available as a part of FragPipe.

### Evaluating sensitivity and false discovery rates (FDR) using a benchmark dataset

To evaluate the sensitivity and false discovery rates of the entire workflow, we used the dataset from Fröhlich et al.[56], consisting of four conditions: "Lymphnode", "1–25", "1–12", and "1–06". The samples used in the first condition contained peptides from *Homo sapiens* only. The other three conditions have a mixture of peptides from *H. sapiens* and *Escherichia coli*. *E. coli* to *H. sapiens* ratios in "1–25", "1–12", and "1–06" are 1 to 25, 1 to 12, and 1 to 6, respectively. Four conditions have the same *H. sapiens* amount. There are 92 wide-window DIA runs (primary DIA runs), and auxiliary MS files used for spectral library building: six narrow-window GPF-DIA runs, and 20 DDA runs. We refer to this as "benchmark" dataset. We performed two MSFragger-DIA based analyses (Fig. 2): (1) building the spectral library using the primary DIA data

(FP-MSF workflow, see Methods); (2) building the spectral library from all available data, that is, also including the GPF-DIA and DDA data (FP-MSF hybrid workflow, see Methods). DIA-NN, which is available as part of FragPipe, was used to quantify the precursors from the DIA runs using these spectral libraries. The Spectronaut 14, 17, and DIA-NN library-free results were used for comparison. The number of *E. coli* precursors detected in the "Lymphnode" condition (that contained *H. sapiens* proteins only) was used to estimate the false discover proportion (FDP, a.k.a. actual FDR or empirical FDR, see Methods).

We also investigated replacing, when performing targeted quantification in DIA-NN, empirically observed fragments and their intensities in the FragPipe/MSFragger-DIA generated spectral libraries with the in silico predicted values. Note that the list of peptide ions included in the library and their retention time values remained unchanged. Replacing the experimental fragment peaks with the predicted peaks resulted in a lower FDP (Supplementary Data 2). The rationale behind this approach is that, in the case of low-abundance peptides, the experimental fragment peaks frequently appear incomplete, with some peptides exhibiting only a few fragment peaks. Also, low-quality peptides may contain interfering peaks, leading to an increased likelihood of false matches. The incompleteness and interference make it challenging for the target-decoy modeling to accurately distinguish true matches from false ones. By replacing the experimental fragment peaks with the in silico predicted peaks, we address this issue of incompleteness, thus improving the model's performance. This option was enabled for all analyses reported in this work.

Figure 2a presents the upset plot illustrating the quantification of precursors obtained from DIA runs. There is a substantial degree of overlap between the precursors quantified by both the FP-MSF and other workflows. Although Spectronaut 17 reported the highest number of precursors, it raised concerns by detecting many unique precursors that were not detected by any other tools, potentially leading

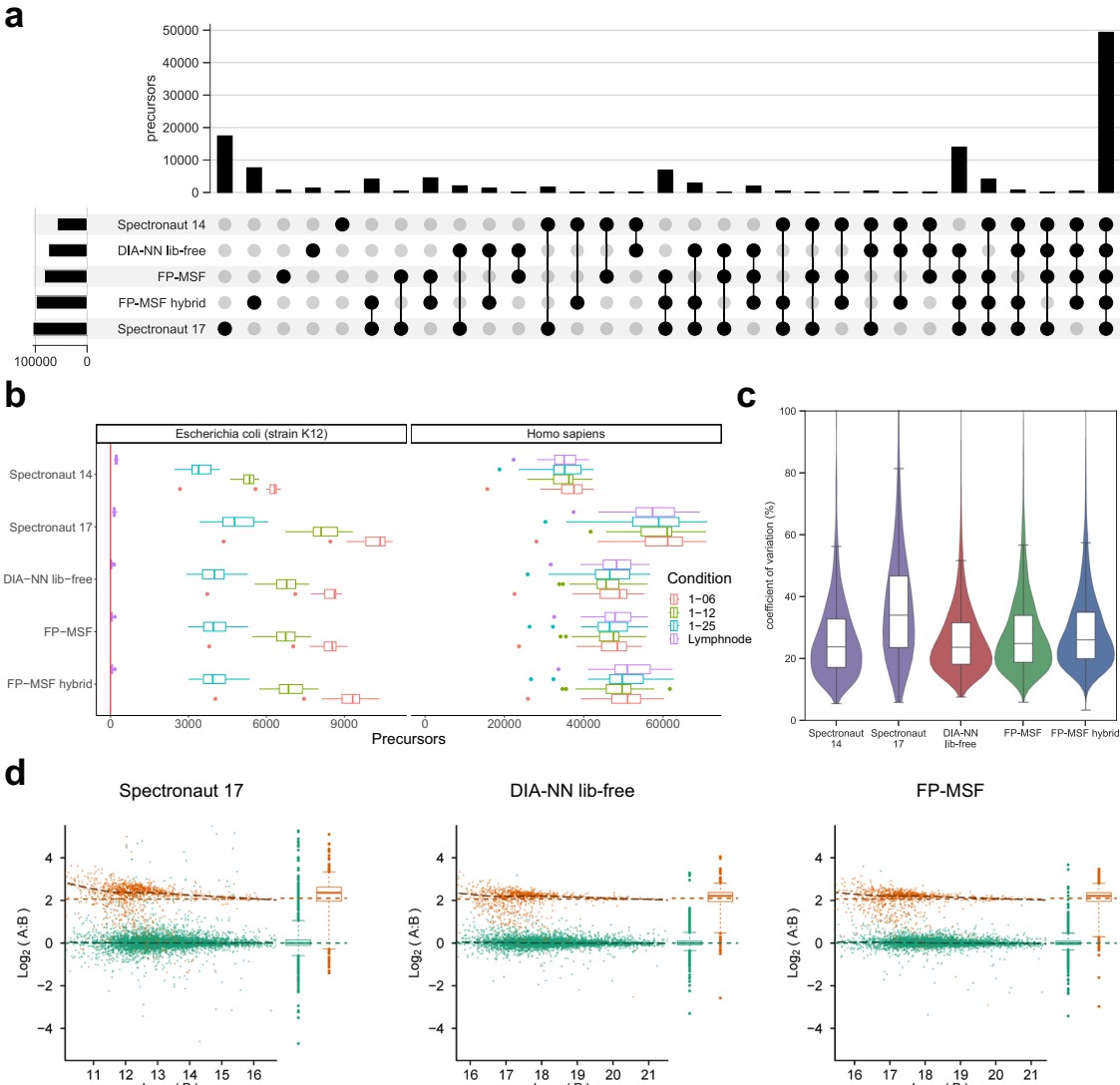

**Fig. 2 | Performance assessment of sensitivity, false discovery rate, precision, and accuracy using a benchmark dataset.** Source data are provided as a Source Data file. **a** Upset plot illustrating the quantified precursors. The precursors are from all four conditions of both *H. sapiens* and *E. coli*. **b** Box plots representing the counts of quantified precursors under four distinct conditions, each with a unique color. There are 4 independent conditions. Each condition consists of 23 single-shot DIA runs from 23 biological independent samples. The "Lymphnode" condition comprises samples from *H. sapiens*, while the remaining three conditions include both *H. sapiens* and *E. coli* spike-in samples. *H. sapiens* and *E. coli* precursors are displayed in separate panels. *E. coli* precursors in the "Lymphnode" condition are deemed false identifications. The box in each box plot captures the interquartile range (IQR), with the bottom and top edges representing the first (Q1) and third quartiles (Q3) respectively. The median (Q2) is marked by a horizontal line within

the box. The whiskers extend to the minima and maxima within 1.5 times the IQR below Q1 or above Q3. Outliers, signified by individual dots, fall outside these bounds. **c** Violin plots showcasing the coefficient of variation (CV) based on *E. coli* precursors from the "1–06" condition. There are 23 replicates. Each violin plot contains an embedded box plot. The box plots' edges, median, and whiskers are same as the previous ones. **d** Scatter plots depicting the relationship between protein log2 ratio and intensity, using proteins from the "1–06" (condition A) and "1–25" (condition B) conditions to compute log-ratios. There are 2 conditions. Each condition contains 23 replicates. *E. coli* proteins are colored orange, while *H. sapiens* proteins appear in green. Horizontal dashed lines indicate true log-ratios, while adjacent box plots display the marginal distribution of log-ratios on the right side of each scatter plot. The box plots' edges, median, and whiskers are same as the previous ones.

to false positives. To assess false discovery rates, we analyzed the number of *E. coli* precursors present in the "Lymphnode" condition, as depicted in Fig. 2b. The *E. coli* box plots show that all tools control the false positives well (all *E. coli* peptide detections in the "Lymphnode" condition are assumed to be false), with slightly higher number of false identifications observed for Spectronaut 14 and 17 (Supplementary Data 2). The box plots from the other three conditions, for both *H. sapiens* and *E. coli* precursors, show that the MSFragger-DIA based workflows and DIA-NN library-free resulted in comparable number of precursors. Including DDA and GPF-DIA data in the spectral library building step (FP-MSF hybrid workflow) provided an additional small boost in the number of quantified precursors.

## Quantification precision and accuracy evaluation

We used the same "benchmark" dataset to evaluate the precision and accuracy of DIA data quantification. Since the 23 runs in the same condition have the same amount of *E. coli* spike-in peptides, we calculated the coefficient of variation (CV) for each *E. coli* precursors in the "1–06" condition. We compared our tools with Spectronaut 14, 17, and DIA-NN library-free pipelines. Figure 2c shows the violin plot of the CV distribution, with box plots illustrating the quartiles and median of the CVs. The plots reveal that, except for Spectronaut 17, the tools have comparable quantification precision. Spectronaut 17 exhibits the noticeably high CVs, indicating the lowest precision among all tools.

We further assessed the quantification accuracy using proteins from the "1−06" and "1−25" conditions. Between these conditions, the *H. sapiens* proteins maintain the same quantity, while the *E. coli* proteins have a 25 to 6 ratio. Figure 2d and Supplementary Fig. 2 show scatter plots of the log-ratios versus the intensities in both protein and precursor levels. Green dots represent *H. sapiens* proteins and orange dots indicate *E. coli* proteins. The horizontal dashed lines signify the true log-ratios, while the curved dashed lines indicate the trend of the dots. Box plots on the right of each scatter plot demonstrate the marginal distribution of the log-ratios. The figures reveal that FP-MSF, FP-MSF hybrid, and DIA-NN library-free have comparable accuracy. However, Spectronaut 14 and 17 exhibit lower quantification accuracy due to noisier scatter plots and a higher number of outliers. This analysis demonstrates that MSFragger-DIA in conjunction with Frag-Pipe offers high precision and accuracy in quantifying DIA data, while also maintaining low false discovery rates and comparable sensitivity among state-of-the-art tools.

### Staggered-windows DIA with gas phase fractionation (GPF)

We then tested the ability of MSFragger-DIA to identify peptides from DIA data acquired using the "staggered" windows approach[57,58], with additional narrow-window GPF-DIA data acquired for spectral library building. The first dataset, taken from Searle et al.[13], contains six narrow -window GPF-DIA runs plus three wide-window, single-injection DIA runs (the primary DIA runs for quantification) from a HeLa lysate, 130 min gradient time. The isolation windows of the GPF-DIA and single-injection DIA runs are 4 Th and 24 Th, respectively. After demultiplexing the staggered windows using ProteoWizard[59], the effective isolation windows were halved. We refer to this dataset as "2018-HeLa". The second, similar dataset was taken from Searle et al.[60] and contains six GPF-DIA runs, and four single-injection DIA runs of *Saccharomyces cerevisiae* lysate,115 min gradient time. The isolation windows are 4 Th and 8 Th, respectively, with effective widths halved after demultiplexing. We refer to this dataset as "2020-Yeast". More information regarding these data can be found in Supplementary Data 1.

We used the FP-MSF workflow to analyze these two datasets (in each dataset, GPF-DIA and primary DIA data were processed together using MSFragger-DIA). We also compared the results with that from the original publication[60], and with the results of running Spectronaut 17, EncyclopeDIA, and DIA-NN library-free pipelines (see Methods). Figure 3a and Supplementary Fig. 3a show the numbers of quantified peptides from single-injection DIA runs. Although the 2018-HeLa

dataset was published earlier[13], Searle et al.[60] re-analyzed the same dataset using an optimized spectral library. Thus, we included the results from Searle et al.[60] in the Figure (labeled "Searle et al. 2020"). Bar heights indicate the average number of quantified peptides, and circles show the number of quantified peptides in each individual run. Spectronaut 17 demonstrated the highest sensitivity among the compared tools. To examine if those quantified peptides are true and have high quality, we generated box plots to show the distributions of CVs in Fig. 3b and Supplementary Fig. 3b. The blue ones are from the peptides overlapped among all tools, and the brown ones are the unique peptides from a specific tool. No CVs data were available from Searle et al.[60]. Even though Spectronaut 17 quantified more peptides, the quantification precision is much lower (high CVs) compared to the peptides from other tools. EncyclopeDIA has the second highest CVs for its unique peptides. DIA-NN library-free and FP-MSF (which also uses DIA-NN, but for quantification only using the FP-MSF generated library) yielded similarly low CVs. Overall, this analysis shows that MSFragger-DIA works well with GPF-DIA and staggered window DIA data, exhibiting both good precision and sensitivity.

### Low-input and single-cell proteomics data

We then used a dataset published by Siyal et al.[16] to demonstrate the performance of MSFragger-DIA in analyzing low-input cell data. We selected two experiments containing 0.75 ng and 7.5 ng of starting material. Each experiment was performed in three replicates. A detailed list of these files is provided in Supplementary Data 1. We refer to this dataset as "low-input-cell". The authors also generated DIA data on samples with higher amounts of starting material, 1.5 ng and 1 μg, for the purpose of building spectral libraries. To compare with the published result, we used the FP-MSF pipeline to perform two analyses. The first analysis used the spectral files from the 0.75 ng samples (primary DIA data for quantification) and the 1.5 ng samples (DIA data used for spectral library building only). The second analysis used the spectral files from the 7.5 ng and 1 μg samples. We also used FP-DIAU, DIA-NN library-free, and MaxDIA pipelines to analyze the same data for comparison (see Methods). Figure 4a, b show the number of quantified proteins from 0.75 ng and 7.5 ng samples, respectively. All the tools were run using the same set of input DIA data. We used the MaxLFQ approach[61] for peptide–protein intensity roll-up in all workflows that used DIA-NN quantification (see Methods). However, we used the "Intensity" columns for MaxDIA because of a very high rate of missing quantification values with the "LFQ Intensity" columns (MaxLFQ approach, Supplementary Fig. 4a, b). The figures show that FP-MSF and

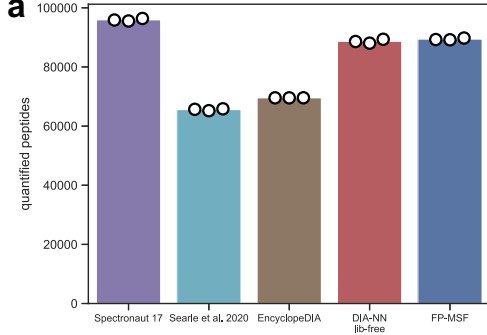

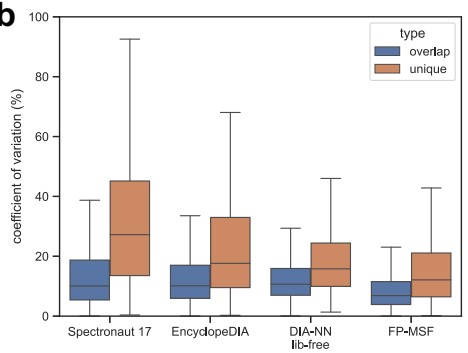

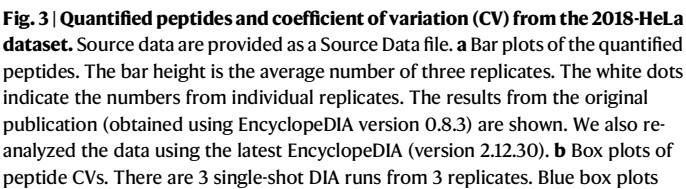

**Fig. 3 | Quantified peptides and coefficient of variation (CV) from the 2018-HeLa dataset.** Source data are provided as a Source Data file. **a** Bar plots of the quantified peptides. The bar height is the average number of three replicates. The white dots indicate the numbers from individual replicates. The results from the original publication (obtained using EncyclopeDIA version 0.8.3) are shown. We also re-analyzed the data using the latest EncyclopeDIA (version 2.12.30). **b** Box plots of peptide CVs. There are 3 single-shot DIA runs from 3 replicates. Blue box plots

represent overlapping peptides shared among all tools, while brown box plots depict unique peptides quantified exclusively by each specific tool. The box in each box plot captures the IQR, with the bottom and top edges representing the Q1 and Q3 respectively. The median is marked by a horizontal line within the box. The whiskers extend to the minima and maxima within 1.5 times the IQR below Q1 or above Q3.

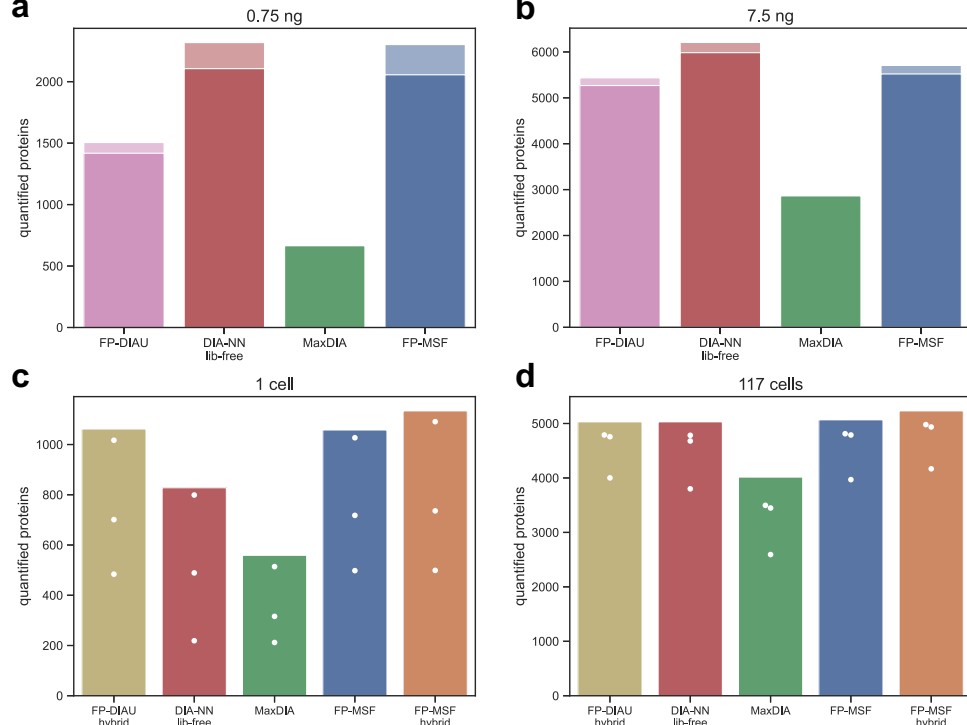

**Fig. 4 | The number of quantified proteins from the low-input-cell and the single-cell datasets.** Source data are provided as a Source Data file. **a**, **b** Bar plots from analyzing the low-input-cell dataset with 0.75 ng and 7.5 ng of starting material. Proteins with missing values (zero intensities) were discarded. The dark color is from the proteins with CVs less than 20%, and the light color is from the proteins with CVs greater than or equal to 20%. **c**, **d** Same as above, for the single-cell dataset, for 1 cell and for 117 cell data.

DIA-NN library-free pipelines have similar sensitivities that are higher than those of the other tools.

We also used the single-cell dataset published by Gebreyesus et al.[15]; we refer to this dataset as "single-cell". We picked two experiments with 1 cell and 117 cells. Each experiment has three replicates. The authors also generated two sets of DIA runs to build two spectral libraries, referred to as "small-sample-lib" and "large-sample-lib" (with the latter generated using a higher amount of starting material, see Methods and Supplementary Data 1). There are 9 DIA runs and 3 DDA runs in the small-sample-lib list of files. There are 4 DIA runs and 4 DDA runs used for large-sample-lib. We used FP-DIAU hybrid, MaxDIA, DIA-NN library-free, FP-MSF, and FP-MSF hybrid (using DDA data in addition to the library-only DIA data) pipelines to analyze this dataset (see Methods). Figure 4c, d show the number of quantified proteins from 1 cell and 117 cells experiments, respectively. As for the low-input-cell dataset, for MaxDIA we used the "Intensity" instead of "LFQ Intensity" because a significant fraction of the proteins had zero MaxLFQ intensity (Supplementary Fig. 4c, d). The results show that FP-MSF hybrid pipeline quantified more proteins than the other tools in the 1-cell experiment. In the 117-cell experiment, the difference between different pipelines was less noticeable.

**Phosphoproteomics data**

After showing that MSFragger-DIA performs well in global proteome data, including single-cell data, we used a phosphopeptide-enriched dataset[62] (Supplementary Data 1) to evaluate the performance of MSFragger-DIA in identifying phosphorylated peptides. The dataset contains two single-injection replicates for six different melanoma cell lines. We refer to this dataset as the "melanoma-phospho" dataset. The data were acquired with variable-width isolation windows over a 120 min gradient. Because no DDA data were generated in this experiment, we used FP-DIAU, DIA-NN library-free, and FP-MSF pipelines (see Methods for details). Proteins with more than 50% of missing

values were discarded. Although all pipelines produced similar numbers of quantified phosphopeptides (Fig. 5a), this dataset highlights the speed advantage of MSFragger-DIA. We used a Windows desktop and a Linux server (see Methods) to compare the computational times (Fig. 5b, c). Prior to analysis, the 12 raw files were converted to mzML format, which took approximately 7 min. For the sake of simplification, we excluded this conversion time from the runtime comparison, as it was a prerequisite for all three tools. The total runtime was broken down into different steps. For FP-DIAU workflow there are pseudo-MS/MS extraction by DIA-Umpire, database search using MSFragger in the DDA mode, rescoring, FDR filtering, and DIA-NN quantification steps. For DIA-NN library-free, there are in silico spectral library prediction, identification, and quantification steps. For the FP-MSF pipeline there are database searching using MSFragger-DIA, rescoring and FDR filtering, and DIA-NN quantification steps. FragPipe with MSFragger-DIA had the fastest overall speed; it is at least six times faster than that of the DIA-NN library-free workflow for these data.

**Large-scale DIA based quantification study**

We used a clear cell renal cell carcinoma (ccRCC) cohort[63] from a recent Clinical Proteomic Tumor Analysis Consortium (CPTAC) project to demonstrate large-scale DIA analysis with MSFragger-DIA. This dataset contains 187 single-injection DIA runs from normal and tumor samples, plus 8 fractionated DDA runs from pooled samples, all acquired with a 140 min LC gradient. Detailed file lists are provided in Supplementary Data 1. We used the FP-DIAU, FP-DIAU hybrid, DIA-NN library-free, MaxDIA, FP-MSF, and FP-MSF hybrid pipelines to analyze the data (see Methods). Figure 6a shows the numbers of proteins quantified from the single-injection DIA runs. Proteins with more than 50% of missing values were discarded. Supplementary Fig. 5 shows the numbers without discarding any proteins. Without using DDA data, FP-DIAU and FP-MSF performed similarly, quantifying ~6500 proteins, followed by DIA-NN library-free and MaxDIA. Adding DDA data as part

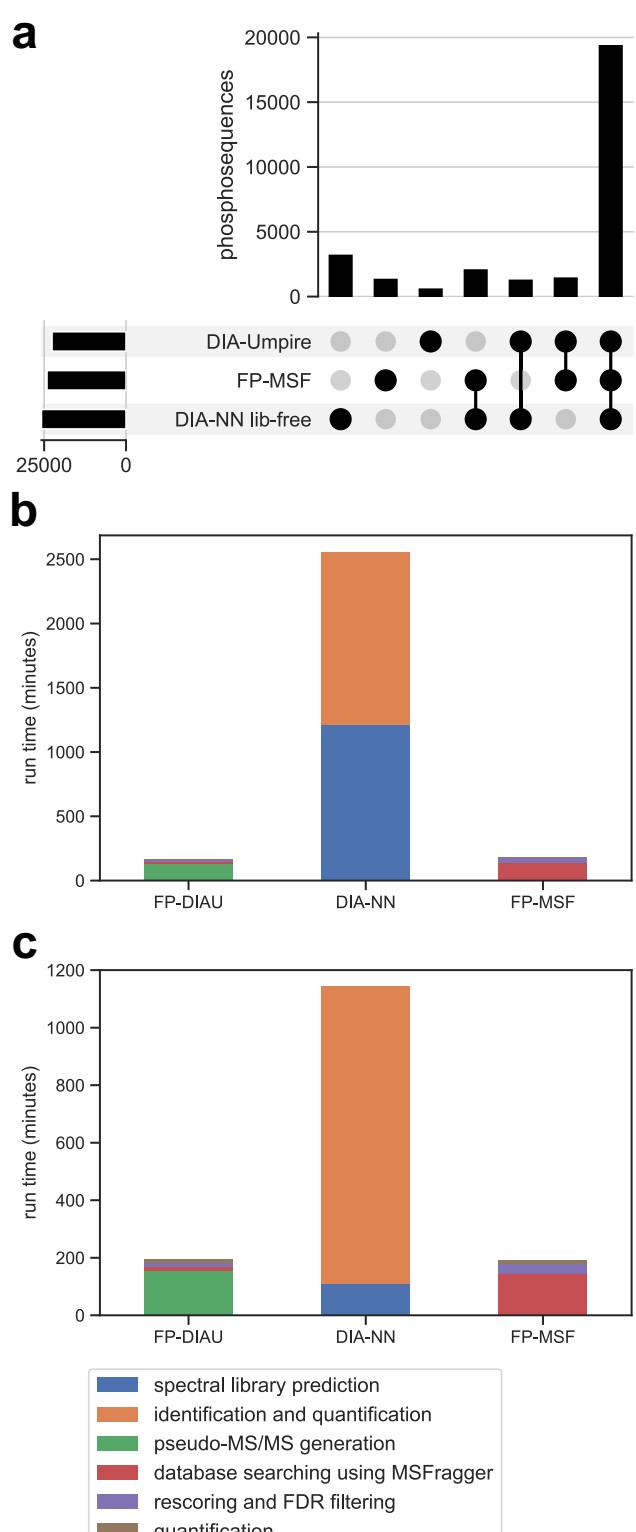

**Fig. 5 | The number of quantified phosphopeptide sequences and runtime for the melanoma-phospho dataset.** Source data are provided as a Source Data file. **a** An upset plot of the number of quantified phosphopeptide sequences. **b** The runtime analysis performed on a Windows desktop. **c** The runtime analysis performed on a Linux server.

of the spectral library building step resulted in a small increase in the identification numbers (FP-DIAU hybrid and FP-MSF hybrid), again confirming the utility of using auxiliary data, such as fractionated DDA data, when available.

Running FP-MSF hybrid pipeline took less than 20 h on the Linux server, and MSFragger-DIA based analyses had the fastest run time. Because it was impractical to time the entire dataset of 187 files uninterruptedly, a more detailed comparison between the pipelines was carried out using a subset of 20 DIA runs. The jobs were run on the same standard Windows desktop (Fig. 6b) and the same Linux server (Fig. 6c), as mentioned in the previous section (see Methods). The time spent on predicting the in silico spectral library for MaxDIA was not included because a previously generated library was used. For FP-DIAU, DIA-NN, and FP-MSF, the raw files were converted to mzML format, which took approximately 8 min. To maintain clarity and simplicity in the figures, we excluded this conversion time from the runtime comparison. FP-MSF pipeline (including the quantification using DIA-NN) was at least two times faster than DIA-NN library-free analysis, and five times (Windows desktop) or ten times (Linux server) faster than MaxDIA. We also compared the runtime between Spectronaut 17 and FP-MSF pipelines on another Windows desktop (see Methods). Spectronaut 17 took 337 min and FP-MSF took 144 min.

We also evaluated the FDR control using the entrapment database approach[64]. The FDP for DIA-NN library-free, FP-MSF, and FP-MSF hybrid pipelines are 1.2%, 1.9%, and 1.8%, respectively (see Methods).

Overall, these results show that FragPipe with MSFragger-DIA for peptide identification directly from DIA data can be used to process large-scale datasets, even with relatively standard desktop hardware. In contrast, other pipelines for DIA analysis are likely to require the use of cloud computing pipelines[65], which can be harder for a typical user to install and deploy.

Given the availability of tandem mass tag (TMT) DDA proteomics data—the main quantitative proteomics platform used by the CPTAC[66]—and RNA-seq data for the same ccRCC patients[63], we performed additional analyses across these data types. We used OmicsEV[67], a recently described tool for quality control and comparative evaluation of omics data tables. We used as input, in addition to the RNA-seq (taken from ref. 63) and TMT-based quantification data (mzML files from ref. 63, reprocessed as part of this work using the default FragPipe TMT workflow), DIA protein quantification tables from FP-DIAU, FP-DIAU hybrid, FP-MSF, FP-MSF hybrid, and DIA-NN library-free pipelines. OmicsEV produces comprehensive visual and quantitative plots that help evaluate data quality of individual data tables and facilitate the comparisons (for the full output from OmicsEV, see Supplementary Data 3). Reassuringly, all quantitative proteomics data tables produced similar results across all OmicsEV metrics. Figure 6d shows the principal component analysis (PCA) plot for the quantification tables from the FP-MSF hybrid pipeline. The tumor and normal samples are well separated, as observed in the original study, based on the TMT DDA data. OmicsEV also calculated the gene-wise correlation between the RNA and protein abundances, with Spearman's correlation for the FP-MSF hybrid pipeline results shown in Fig. 6e. For comparison, the gene-wise correlation between the RNA and TMT-based protein abundance data is shown in Fig. 6f. DIA and TMT DDA-based protein quantification data have similar gene-wise correlations with RNA abundances, albeit slightly higher for the TMT DDA-based data. Furthermore, more proteins were quantified using TMT DDA. This is not unexpected, however, given that the TMT DDA data was generated on highly fractionated peptide samples (25 fractions), whereas DIA data was acquired without fractionation. The functional pathways identified as enriched in tumor vs. normal samples were nevertheless similar between the DIA and TMT-based data (Supplementary Data 3). Overall, our analysis shows that DIA data is comparable to that from the more established, TMT-based protein quantification platform.

## Discussion
We described a new method for identification of peptides directly from DIA data. The MSFragger-DIA strategy differs from that of DIA-

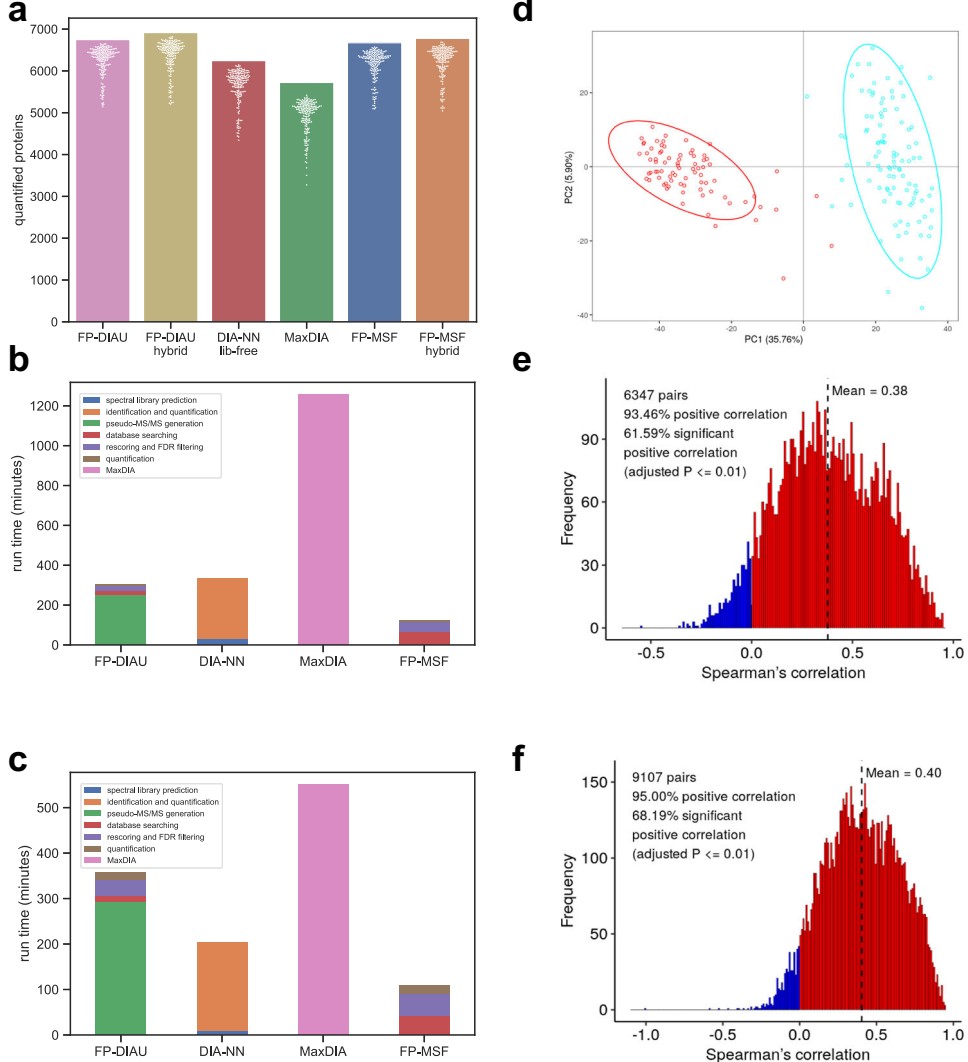

**Fig. 6 | Results from the ccRCC dataset.** Source data are provided as a Source Data file. **a** Bar plots of the number of quantified proteins in the ccRCC dataset. There are 187 independent biological samples. The bar height is the total number. The white dots are the numbers from individual runs. Proteins with more than 50% missing values are discarded. **b** The runtime of analyzing 20 runs of the ccRCC dataset on a Windows desktop. **c** The runtime of analyzing 20 runs on a Linux server. **d** PCA plot based on the FP-MSF hybrid results, showing tumor (blue) and normal (red)

samples. **e** Histogram of Spearman's correlation coefficients between the RNA-Seq and the DIA protein abundance data (FP-MSF hybrid pipeline). The adjusted p-value is from the two-sided test followed by the Benjamini-Hochberg procedure.
**f** Histogram of the Spearman's correlation coefficients between the RNA-Seq and the TMT DDA-based protein abundance data. The adjusted $p$-value is from the two-sided test followed by the Benjamini-Hochberg procedure.

Umpire by reversing the order in which the two key steps are performed: detection and correlation of precursor and fragment ion features, and peptide-spectrum matching. While DIA-Umpire starts with untargeted feature detection, MSFragger-DIA first determines a list of all meritorious candidate peptides for each MS/MS scan without spectral deconvolution. Feature detection and peak tracing across the LC dimension are performed in MSFragger-DIA as a second step, in a targeted manner, and for selected candidate peptide ions. Using several common experimental DIA workflows, we demonstrated that MSFragger-DIA is faster than DIA-Umpire followed by the conventional (DDA mode) MSFragger search. It also has a higher sensitivity for peptide identification, in part due to improved feature detection and peak tracing performed in MSFragger-DIA in the targeted mode. However, we consider the MSFragger-DIA and DIA-Umpire workflows to be complementary. One advantage of the DIA-Umpire is that extraction of pseudo-MS/MS spectra needs to be done once; these spectra can then be re-searched using MSFragger (in DDA mode) multiple times, for example, using different sequence databases or

search parameters. In contrast, MSFragger-DIA needs to be run each time, starting from the raw data. Furthermore, the speed advantage of MSFragger over DIA-Umpire is less significant for searches containing multiple variable modifications. For nonspecific searches (e.g., HLA peptidome data), MSFragger-DIA is significantly slower; thus, the default FragPipe workflow for nonspecific DIA searches is based on DIA-Umpire.

MSFragger-DIA, with its direct database search-first approach, essentially blurs the difference between the analysis of DIA and DDA MS/MS spectra. Thus, we believe that MSFragger-DIA can be further extended to the analysis of wide-window DDA data[68,69], enabling identification of chimeric (co-fragmented) peptides from such data. The strategy described here is also different from those based on peptide-centric searches using in silico predicted spectral libraries. Using MSFragger-DIA, it is not necessary to predict the entire spectral library in advance, and there is no restriction on the size of peptides, peptide ion charge states, or modifications that can be considered. However, our workflows in FragPipe (for both DDA and DIA data) do

benefit from deep learning-based predictions at the subsequent, rescoring stage with the help of MSBooster and Percolator[54]. Interestingly, we observed some advantages of using predicted spectra (instead of empirical ones) at the final targeted quantification extraction step. This indicates that while MSFragger-DIA has high sensitivity for peptide identification, some fragment ions may not be detectable in every MS/MS scan where the peptide was confidently identified. Because the spectral library building step in EasyPQP considers each MS/MS scan in isolation, some fragments that are detectable using targeted peak tracing (in FragPipe, using DIA-NN) may be missing in the empirical spectra in the library. Replacing the empirical spectra with the predicted spectra improves the number of quantified precursors. Such a replacement, however, is not advised in all situations, for example, when performing MSFragger-DIA searches with less common variable modifications, because such spectra may not be predicted well.

Since this study focuses on the peptide identification, we did not thoroughly investigate other aspects of the DIA data analysis such as modification localization. However, by integrating with FragPipe and DIA-NN, we can perform site localization in the DIA-NN quantification step[5]. Lou et al.[70] demonstrated that the DIA-NN localization algorithm was more conservative than Spectronaut's but resulted in lower sensitivity. Considering the modification localization for the DIA data is still an open question, we will investigate it in the future work.

MSFragger coupled with FragPipe enables researchers to conduct DIA data analysis using different approaches, including library-based, library-free, and hybrid methods. The term "library" in this context typically refers to DDA or GPF-DIA data obtained from fractionated pooled samples. An in silico spectral library from deep-learning predictions is not considered a "library", as it eliminates the need for extra sample preparation and data acquisition. Within the FragPipe, the library-based approach employs DDA or GPF-DIA data to create a spectral library. The library-free approach directly searches DIA data to generate a library. The hybrid approach combines DDA/GPF-DIA and DIA data to construct a spectral library. The library is then used as a reference for targeted extraction and identification of peptides in DIA data. This method utilizes the strengths of both library-based and library-free methods, resulting in a more comprehensive library. Our experiments have shown that the hybrid approach tends to result in the quantification of more peptides and proteins in comparison to library-based and library-free methods. One of the reasons is the enhanced library coverage. By integrating both DDA/GPF-DIA and DIA data, the hybrid approach generates a spectral library with a broader range of peptide and protein entries. This increased coverage allows for the identification of peptides that may not be present or detectable in either library-based or library-free approaches alone. The deeply fractionated DDA or GPF-DIA data can contribute low-abundance peptides, and the DIA data can contribute unique peptides specific to the experiment. Therefore, we propose to use the hybrid approach when DDA or GPF-DIA data is available.

Recently, Bruker developed a timsTOF platform to couple ion mobility separation with time-of-flight (TOF) mass spectrometer. To generate DIA data using the timsTOF platform, researchers proposed diaPASEF data acquisition strategy[71], and demonstrated a very promising performance of this platform. diaPASEF data poses new challenges to data analysis, as it becomes necessary to accommodate an additional dimension of ion mobility separation. To address some of those challenges, researchers have been investigating different approaches to generate the data. Three protocols, including Slice-PASEF[72], midiaPASEF[73], and Synchro-PASEF[74], have been proposed very recently. Due to the new data structure and the rapid development, support in MSF-DIA for diaPASEF data is beyond the scope of this manuscript. In future work, we plan to leverage the fragment ion indexing to support direct peptide identification from diaPASEF data in MSFragger.

In summary, we have developed a new direct DIA peptide identification method and implemented it as a module of the MSFragger search engine. By integrating MSFragger-DIA into the FragPipe computational platform, users can perform one-stop DIA data analysis, from peptide identification to quantification, optionally with the use of auxiliary (e.g., DDA) data to achieve optimal performance. With experiments covering various data acquisition schemes and sample types, we show that MSFragger-DIA demonstrates high sensitivity, fastest speed, and comparable quantification precision to other state-of-the-art DIA analysis tools. Coupled with the ease of use of FragPipe, we believe that this work describes an attractive computational solution for the analysis of a wide range of DIA datasets.

## Methods

### DIA data analysis pipelines
MSFragger and FragPipe can perform one-stop DIA data analysis with and without the use of auxiliary DDA data to build the library. The comparisons were done with the DIA-Umpire workflow in FragPipe based on pseudo-MS/MS generation, Spectronaut directDIA (library-free) analysis, DIA-NN library-free analysis (based on in silico spectral library prediction), EncyclopeDIA in silico library-based analysis, and MaxDIA in silico library-based analyses. We used DIA-Umpire (version 2.2.8), Spectronaut (version 14 and 17, Biognosys), DIA-NN (version 1.8.1), EncyclopeDIA (version 2.12.30), MaxDIA in MaxQuant (MaxQuant version 2.1.3.0 for the runtime comparison and 2.2.0.0 for other experiments), and MSFragger (version 3.5). The pipelines are briefly described below.

FP-MSF: Only DIA data were used in this pipeline. MSFragger-DIA was used to directly search the DIA data. The search results were processed using MSBooster for deep learning-based score calculation, Percolator[33] for rescoring and posterior error probability calculation, ProteinProphet[34] for protein inference, Philosopher[55] for FDR filtering, and EasyPQP for spectral library building. The peptide ions in the spectral library were filtered with 1% global peptide and protein FDR. The resulting library was passed to DIA-NN[23] to extract and quantify precursors, peptides, and proteins from the DIA data. For peptide–protein quantification roll-up, MaxLFQ normalization was performed using the R package available at https://github.com/tvpham/iq.

FP-MSF hybrid: Both DIA and DDA data were used in this pipeline. MSFragger in the DIA and DDA modes was used to search the DIA and DDA data, respectively. The subsequent steps are the same as those in the FP-MSF pipeline.

FP-DIAU: Only DIA data was used in this pipeline. DIA-Umpire was used to generate pseudo-MS/MS spectra from the DIA data. Then, MSFragger in DDA mode was used to search these spectra. The remaining steps are the same as those used in the FP-MSF pipeline.

FP-DIAU hybrid: Both DIA and DDA data were used in this pipeline. MSFragger in the DDA mode was used to search the DIA extracted pseudo-MS/MS spectra and DDA data. The remaining steps are the same as those used in the FP-MSF pipeline.

Spectronaut: Only DIA data was used in this pipeline. The direct-DIA from Spectronaut 14 and directDIA+ from Spectronaut 17 were used to analyze the data. Due to the commercial license, our access to the Spectronaut software is limited. Therefore, only a few experiments use Spectronaut.

DIA-NN library-free: Only DIA data was used in this pipeline. DIA-NN predicted an in silico spectral library from the protein sequence database and then used the library to search and quantify the DIA data. This mode of DIA-NN is also known as DIA-NN library-free mode. MaxLFQ normalization was used for peptide–protein quantification roll-up.

MaxDIA: Only DIA data was used in this pipeline. MaxDIA[49] inside MaxQuant was used to analyze the DIA data using the in silico spectral library downloaded from https://datashare.biochem.mpg.de/s/

qe1IqcKbz2j2Ruf?path=%2FDiscoveryLibraries. This is also known as the "MaxDIA discovery mode"[49]. Unless otherwise noted, MaxLFQ intensity was used. Due to the limitation of the in silico spectral library, experiments involving multiple species or phosphoproteomics do not use MaxDIA.

EncyclopeDIA: This pipeline requires GPF-DIA data to build an experiment-specific spectral library to analyze the single-injection DIA data. Since not all datasets contain GPF-DIA data, EncyclopeDIA was only used in some of the experiments. Prosit[40] (https://www.proteomicsdb.org/prosit/) was used to predict the in silico spectral library from a provided fasta file. Subsequently, EncyclopeDIA used six GPF-DIA runs to generate an experiment-specific spectral library by searching against the in silico spectral library. This experiment-specific spectral library was used to analyze the single-injection DIA runs with EncyclopeDIA. These steps were recommended by Searle et al.[60] and detailed in the supplementary documents of the original publication.

To ensure a fair comparison, we used the iq R package[75] to filter the results and calculate the MaxLFQ[61] intensity for FP-MSF, FP-MSF hybrid, FP-DIAU, FP-DIAU hybrid, Spectronaut, and DIA-NN library-free pipelines. MaxDIA and EncyclopeDIA pipelines are not compatible with the iq package; hence, their results were filtered using the tools themselves. For FP-MSF, FP-MSF hybrid, FP-DIAU, FP-DIAU hybrid, DIA-NN library-free, and EncyclopeDIA pipelines, the input data files were converted to mzML format using ProteoWizard[76] to facilitate a fair and straightforward comparison. However, Spectronaut and MaxDIA perform better with raw format, so we used raw files for those two pipelines. It is worth noting that both FP-MSF and FP-MSF hybrid pipeline support raw format on Windows operating system. Converting to mzML format is not mandatory. Additionally, there is no need to run the iq R package in real applications, as our tools can carry out FDR filtering and MaxLFQ intensity calculation.

## Sensitivity and FDR evaluation

The dataset published by Fröhlich et al.[56] was used to benchmark the sensitivity and estimate the false discover proportion (FDP, a.k.a. actual FDR or empirical FDR). There are 92 single-injection DIA runs, 6 GPF-DIA runs, and 20 DDA runs. Among single-injection DIA runs, there are four conditions: "Lymphnode", "1–25", "1–12", and "1–06". Each condition has 23 runs from different patients. Details of sample preparation and data acquisition can be found from the original publication. We used the DIA runs in the Spectronaut, DIA-NN library-free, and FP-MSF pipelines. We used all the DIA, GPF-DIA, and DDA runs in the FP-MSF hybrid analysis. The FASTA file was a combination of *H. sapiens*, *E. coli*, and common contaminant proteins (downloaded on February 18, 2022, UP000005640 and UP000000625, 24750 proteins). The enzyme was set to restricted trypsin (i.e., allowing cleavage before Proline). Carbamidomethyl cysteine was set as a fixed modification. Protein N-terminal acetylation and oxidation of methionine were set as the variable modifications. The maximum allowed number of missed cleavages was set to 1. For DIA-NN library-free, FP-MSF, and FP-MSF hybrid, the precursors were filtered with the combination of 1% run-specific precursor FDR, global precursor FDR, run-specific protein FDR, and global protein FDR. For Spectronaut 14, the precursors were filtered with the combination of 1% precursor FDR and global protein FDR. There is no run-specific protein FDR available. For Spectronaut 17, the precursors were filtered with the combination of 1% precursor FDR, run-specific protein FDR, and global protein FDR. The Zenodo link to the detailed parameters can be found in the "Data availability" section.

There are only *H. sapiens* peptides in the "Lymphnode" experiment. Thus, the FDP can be calculated as:

$$\text{FDP} = \frac{n_{ft}}{n_t} \quad (1)$$

where $n_{ft}$ is the number of false *H. sapiens* peptides and $n_t$ is the total number of detected *H. sapiens* peptides. To estimate the FDP, we first have:

$$\frac{n_d}{N_d} \approx \frac{n_{ft}}{(N_t - n_{tt})} \quad (2)$$

where $n_d$ is the number of detected *E. coli* peptides, $N_d$ is the number of *E. coli* peptides in the spectral library, $N_t$ is the number of *H. sapiens* peptides in the spectral library, and $n_{tt}$ is the number of detected true *H. sapiens* peptides. Combining Eqs. (1) and (2), and assuming $n_{tt} \approx n_t$, we have:

$$\text{FDP} \approx \frac{n_d}{n_t} \cdot \frac{(N_t - n_{tt})}{N_d} \approx \frac{n_d}{n_t} \cdot \frac{(N_t - n_t)}{N_d} \quad (3)$$

## Quantification precision and accuracy evaluation

We used the same dataset[56] used in the previous section to evaluate the precision and accuracy of quantification. In the "1–06" condition, each run contains an equal amount of *E. coli* spike-in peptides. Those peptides were used to calculate CV for benchmarking the precisions of the tools. Between the "1–06" and "1–25" conditions, the *H. sapiens* proteins maintain the same quantity, while the *E. coli* proteins are expected to have a 25 to 6 ratio. Therefore, we used the proteins intensities from these conditions to evaluate the quantification accuracy. LFQbench[77] was used to generate the scatter plots. In those two evaluations, Spectronaut 14, 17, DIA-NN library-free, FP-MSF, and FP-MSF hybrid pipelines were used. The database and parameters are the same as the previous section.

## Staggered isolation window data

We used two sets of DIA data acquired with staggered isolation windows to demonstrate the performance of MSFragger-DIA. The first dataset was published by Searle et al.[13] from HeLa samples acquired on a Thermo Q Exactive HF mass spectrometer. It contains six GPF-DIA runs with 4 Th isolation windows, and three single-injection DIA runs with 24 Th isolation windows. After demultiplexing[59], the effective isolation windows were 2 Th and 12 Th, respectively. We refer to this dataset as 2018-HeLa. The second dataset was published by Searle et al.[60] from *S. cerevisiae* strain BY4741 samples acquired on a Thermo Fusion Lumos mass spectrometer. It contains six GPF-DIA runs with 4 Th isolation windows and four single-injection DIA runs with 8 Th isolation windows. After demultiplexing, the effective isolation windows are 2 Th and 4 Th, respectively. We refer to this dataset as 2020-Yeast. The raw files were converted to mzML format using ProteoWizard[76] with the vendor's peak picking and demultiplexing. Details of the sample preparation and data acquisition can be found in the original publications[13,60].

Spectronaut 17, EncyclopeDIA, DIA-NN library-free, and FP-MSF pipelines were used. The enzyme was set to restricted trypsin. Carbamidomethyl cysteine was set as a fixed modification. Protein N-terminal acetylation and oxidation of methionine were set as the variable modifications. The maximum allowed number of missed cleavages was set to 1. For Spectronaut 17, EncyclopeDIA, DIA-NN library-free, and FP-MSF pipelines, reviewed *H. sapiens* proteins and common contaminant sequences were downloaded from UniProt (downloaded on March 23, 2021, UP000005640, 20431 proteins) for 2018-HeLa dataset. The reviewed *S. cerevisiae* proteins and common contaminants were downloaded from UniProt (downloaded on March 16, 2021, UP000002311, 6164 proteins) for 2020-Yeast dataset. Decoy (reversed) sequences were appended to the original database for MSFragger-DIA analysis. For Spectronaut 17, the quantified peptides were filtered with the combination of 1% precursor FDR, run-specific protein FDR, and global protein FDR. For EncyclopeDIA, the quantified

peptides were filtered with 1% global precursor FDR by the tool itself. For DIA-NN library-free and FP-MSF pipelines, the quantified peptides were filtered with the combination of 1% run-specific precursor FDR, run-specific protein FDR, global precursor FDR, and global protein FDR. The Zenodo link to the detailed parameters can be found in the "Data availability" section.

### Low-input-cell data

A dataset from Siyal et al.[16] was used to demonstrate the performance of MSFragger-DIA in analyzing low-input data. We picked two experiments with 0.75 ng and 7.5 ng cells. Each experiment has three replicates. The authors also generated two sets of auxiliary DIA runs ("library runs") for spectral library building, using samples with 1.5 ng and 1 μg of starting material. The database was obtained from the original publication (20387 proteins). FP-DIAU, DIA-NN library-free, MaxDIA, and FP-MSF pipelines were used because no DDA data was available. For MaxDIA, the *H. sapiens* spectral library named "missed_cleavages_1" from https://datashare.biochem.mpg.de/s/qe1IqcKbz2j2Ruf?path=%2FDiscoveryLibraries was used. For FP-DIAU and FP-MSF workflows, the primary DIA data from the low-input cell runs and library DIA runs were used together to build a spectral library. For FP-DIAU, DIA-NN library-free, and FP-MSF pipelines, the proteins were filtered with the combination of 1% run-specific precursor FDR, run-specific protein FDR, global precursor FDR, and global protein FDR. For MaxDIA, proteins were filtered with 1% global PSM and protein FDR. The remaining parameters are the same as those described in the preceding section.

### Single-cell data

A single-cell dataset published by Gebreyesus et al.[15] was used in this study. We picked two experiments, with 1 and 117 cells. Each experiment has three replicates. The authors also generated two sets of auxiliary runs for spectral library building. The first one has 9 DIA runs and 3 DDA runs from low-input data. We refer to this library as small-sample-lib. The second one has 4 DIA runs and 4 DDA runs from the bulk cells. We refer to it as large-sample-lib. For the 1 cell experiment, we used the small-sample-lib, as suggested by the authors. For the 117 cells experiment, we used the large-sample-lib. The protein sequence database was obtained from the original publication (20194 proteins). FP-DIAU hybrid, DIA-NN library-free, MaxDIA, FP-MSF, and FP-MSF hybrid pipelines were used. For the FP-DIAU and FP-MSF hybrid pipelines, both DIA and DDA data from the single-cell and library runs were used to generate a spectral library. For the DIA-NN library-free, FP-MSF, and MaxDIA pipelines, the DIA data from the single-cell and library runs were used. The remaining parameters are the same as above.

### Phosphoproteome data

We used a phosphoproteome dataset obtained from Gao et al.[62]. There are six experiments from six melanoma cell lines. Each experiment has two replicates. FP-DIAU, DIA-NN library-free, and FP-MSF pipelines were used. MaxDIA was not used to analyze these data because no phosphoproteome spectral library was available on the corresponding tool website. The enzyme was set to restricted trypsin. Carbamidomethyl cysteine was set as a fixed modification. Protein N-terminal acetylation, oxidation of methionine, and phosphorylation of serine, threonine, and tyrosine were set as variable modifications. The maximum allowed number of missed cleavage was set to 1. The *H. sapiens* proteins and common contaminant sequences downloaded from UniProt (downloaded on March 23, 2021, UP000005640, 20431 proteins) were used as the target protein sequence database. The remaining parameters are the same as above.

### Clear cell renal cell carcinoma (ccRCC) data

In the clear cell renal cell carcinoma (ccRCC) cohort[63], there are 187 single-injection DIA runs from normal and tumor tissues, and 8 fractionated DDA runs from pooled tissues generated for spectral library building. We used FP-DIAU, FP-DIAU hybrid, DIA-NN library-free, FP-MSF, and FP-MSF hybrid pipelines to analyze the data. The enzyme was set to restricted trypsin. Carbamidomethyl cysteine was set as a fixed modification. Protein N-terminal acetylation and oxidation of methionine were set as variable modifications. The maximum allowed number of missed cleavages was set to 1. The database contains the reviewed *H. sapiens* proteins and common contaminant sequences downloaded from UniProt (downloaded on March 23, 2021, UP000005640, 20431 proteins). The Zenodo link to the detailed parameters can be found in the "Data availability" section.

### False discovery rate evaluation using entrapment database

We used 20 DIA runs from the ccRCC cohort[63] to evaluate the FDR control with the entrapment database approach[64]. To ensure a sufficient number of entrapment sequences, we combined databases of *H. sapiens*, *S. cerevisiae*, *E. coli*, and *Arabidopsis thaliana*, obtained from UniProt (downloaded on April 18, 2023, UP000005640, UP000002311, UP000006548, and UP000000625, 47092 proteins). This includes 20407 *H. sapiens* proteins, 6060 *S. cerevisiae* proteins, 4401 *E. coli* proteins, and 16224 *A. thaliana* proteins. We analyzed 20 DIA runs using DIA-NN library-free, FP-MSF, and FP-MSF hybrid pipelines with the custom fasta file. Quantified *H. sapiens* proteins were considered true, while proteins from other species were considered false. We calculated the FDP using the following equation:

$$FDP = \frac{N_h}{N_o} \cdot \frac{n_o}{n_h} \tag{4}$$

where $N_h$ is the number of *H. sapiens* proteins in the database, $N_o$ is the number of non-*H. sapiens* proteins in the database, $n_o$ is the number of quantified non-*H. sapiens* proteins in the results, and $n_h$ is the number of quantified *H. sapiens* proteins in the results.

### Speed benchmarks

The runtime for each pipeline was measured on three computers: (1) Windows desktop: Intel Core i9-10900K, 3.70 GHz, 10 cores, 20 logical processors, 128 GB of memory; (2) Linux server: Intel Xeon E5-2690 v4, 2.6 GHz, 28 cores, 56 logical processors, and 512 GB of memory. We used 19 threads on Windows desktop and 56 threads on the Linux server. And the third computer with a Windows desktop: Intel Xeon W-2175, 2.50 GHz, 14 cores, 28 logical processors, 128 GB of memory. The third computer only ran Spectronaut 17 and FP-MSF pipelines to get the runtime comparison.

### Reporting summary

Further information on research design is available in the Nature Portfolio Reporting Summary linked to this article.

## Data availability

The parameter, log, and result files in this study are available in https://doi.org/10.5281/zenodo.7261712. The raw mass spectrometry data used in this study are available in the MassIVE under accession code MSV000082805 and MSV000084000; in the ProteomeXchange under accession code PXD022992, PXD023325, and PXD027679; in the National Cancer Institute Proteomic Data Commons under accession code PDC000200; and in the European Genome-phenome Archive under accession code EGAD00010002223. The proteome database files used in this study are available in the UniProt database under the proteome ID UP000005640 (*H. sapiens*), UP000002311 (*S. cerevisiae*), UP000000625 (*E. coli*), and UP000006548 (*A. thaliana*). The contaminant protein database used in this study are available in https://www.thegpm.org/crap/. Source data are provided with this paper.

## Code availability

MSFragger-DIA and MSFragger can be downloaded as a single JAR binary file at https://msfragger.nesvilab.org/. FragPipe is available on GitHub at https://github.com/Nesvilab/FragPipe. The Python and R scripts for summarizing the results and generating the figures is available at https://github.com/Nesvilab/MSFragger-DIA-manuscript.

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

## Acknowledgements

This work was funded in part by NIH grants R01-GM-094231, U24-CA210967, U24-CA271037 received by F.Y., G.C.T., A.T.K., G.X.L., A.I.N. This work was also funded in part by German Ministry of Education and Research (BMBF), as part of the National Research Node "Mass spectrometry in Systems Medicine" (MSCoreSys), under grant agreement 161L0221 received by V.D. We thank Sarah Haynes for help with the manuscript, George Rosenberger for help with EasyPQP, and Kai Li for adopting the PDV[78] viewer in FragPipe to support MSFragger-DIA output. We also thank Michael MacCoss for the helpful discussions, and Brian Searle for the help with EncyclopeDIA.

## Author contributions

F.Y. developed MSFragger-DIA, A.T.K. contributed to the algorithm and software development at an early stage of the project. G.C.T. made the FragPipe support Percolator and EasyPQP. V.D. modified DIA-NN for FragPipe and contributed to data analysis. F.Y. and A.I.N. analyzed the data. K.E.F. helped the data analysis using Spectronaut. G.X.L. helped the data analysis using OmicsEV. F.Y. and A.I.N. wrote the manuscript with input from all authors. A.I.N. conceived and supervised the project.

## Competing interests

The authors declare no competing interests.
