## [Peer Review File · Nature Communications]

REVIEWER COMMENTS

Reviewer #1 (Remarks to the Author):

In the work, the group of Prof. Nesvizhskii reports MSFragger-DIA for DIA proteomic data analysis. MSFragger-DIA starts with a direct search of MS/MS spectra against the entire protein sequence database to generate a list of peptide candidates. Then, MSFragger-DIA traces peaks, extracts ion chromatograms, and detects features of all candidate peptides for each spectrum determined. Finally, MSFragger-DIA generates output files compatible with PeptideProphet and Percolator for rescoring and FDR estimation. In general, it proposes a qualification and quantification pipeline for DIA data analysis, supporting DIA analysis without DDA, phosphoproteomics, single cell proteomics, etc. However, the current benchmarking doesn't demonstrate the superior performance of MSFragger-DIA compared to the state-of-the-art software solutions. It is also not clear what is the main innovation point of the current work. Is it just to provide another software solution or are there key advances of the pipeline? The most concept of the current work, including fragment ion indexing, has been proposed by the group or others in the field. The specific comments:

1. There is a lack of comprehensive comparison with the current state-of-the-art software solutions. For comparison, it is necessary to show coverage, reproducibility, quantification accuracy and computation time. It is also necessary to show the consistency among results, i.e. how many proteins shared, how many gain, and how many lose? However, in the current work, the authors show only one or two aspects per dataset comparing to one or two other software solutions, which is not comprehensive enough.
2. For speed comparison, the MSFragger-DIA starts from mzml while the others start from raw, which is not a fair comparison.
3. The authors name MSFragger-DIA as a one-stop analysis pipeline, but it needs DIA-NN for quantification, R packages for normalization and ProteoWizard for data conversion and demultiplexing, which is not one-stop at all.
4. Comparison with spectronaut, the authors compared to different version of spectronaut, from version 13 to version 15. These versions are quite old. The newest version of spectronaut 16 with directDIA2.0 performs much better than these versions. The current comparison is miss leading.
5. The current work doesn't show any benchmarking on timsTOF pro data, which is nowadays a main platform for proteomics research, and should be supported by the software.
6. Comparison with MaxDIA: it mentions that on one dataset, MaxDIA shows low numbers of quantified peptides and on the other dataset, MaxDIA shows a very high rate of missing quantification values. Is it common on different datasets? or is it just specific to the two datasets? What can be the possible reason?
7. Comparison with DIA-NN: we tested DIA-NN and MSFragger-DIA on the dataset ccRCC, which is used for generating Figure 6. Without missing value filtering, DIA-NN in silico library identified 9585 unique genes while FP-MSF identified 8317 unique genes in our result. Even after 50% missing value filtering, DIA-NN in silico library identified 7700 protein groups, 10% more than that shows in Figure 6. In our results, the missing values by DIA-NN and FP-MSF are comparable. Considering computation time, including that for data conversation, the computation time by FP-MSF is faster than DIA-NN in silico library but not significant when analyzing all the 187 files. Our result is not consistent with that in the paper.
8. FDR evaluation, it is suggested to test the FDR using the cohort data (ccRCC) and using entrapment sequences, e.g. E. coli, yeast, A. thaliana. The current benchmarking is not enough. It mentions that "Replacing the experimental fragment peaks with the predicted peaks resulted in a lower actual FDR (Supplementary Table S2)." What is the reason? Normally, experimental library is more specific to the DIA data and should perform better.
9. Single cell proteomics, the current protein identification number is lower than many state-of-the-art

reports (<https://doi.org/10.1101/2022.06.28.498038>), which of course is related to the dataset. The authors should choose timsTOF pro datasets for test, where > 2000 or even 3000 proteins were reported for single cell proteomics.

10. Phosphoproteomics, synthetic phosphopeptides dataset (e.g. Nature Communications 2020, 11, 787) should be adopted to demonstrate the false localization rate of phosphate. For phosphoproteomics, it should be compared to spectronaut instead of DIA-NN, where the latter is not yet optimized for phosphoproteomics.

11. Different methods were suggested for MSFragger-DIA, those with and without DDA data library, and with in-silico library. It should be compared the identification consistency and quantification accuracy of the different methods, and it should be discussed which one is recommended under what condition.

Minor: gas phase fractionation (GFP) should be GPF.

Reviewer #2 (Remarks to the Author):

The reviewer would like to answer the following questions:

What are the noteworthy results?

A fast and sensitive platform for DIA identification and quantification.

Will the work be of significance to the field and related fields? How does it compare to the established literature? If the work is not original, please provide relevant references.

Yes, the work is of significance. Comparing to in silico library-based DIA-NN, MSFragger-DIA is faster, and has similar or better DIA quantification results.

Does the work support the conclusions and claims, or is additional evidence needed?

Yes, it does.

Are there any flaws in the data analysis, interpretation and conclusions? Do these prohibit publication or require revision?

Yes, please see the below comments for minor revision.

Is the methodology sound? Does the work meet the expected standards in your field?

Yes, it sounds good, and meet the expected standards.

Is there enough detail provided in the methods for the work to be reproduced?

Yes, it is.

The authors described the algorithm of MSFragger-DIA, compared to other tools (DIA-Umpire, Spectronaut, DIA-NN, MaxDIA), demonstrated the advantage of MSFragger-DIA (faster, similar or better than DIA-NN). The way of building libraries and the pipeline for identification make MSFragger-DIA different from other tools. If the authors can follow the minor suggestions, the manuscript will become better.

Here are the minor suggestions.

1. Figure 2 shows FP-MSF and FP-MSF hybrid outperform DIA-NN and Spectronaut at 1% FDR. How about different FDR cutoff (e.g. 0.5%, 5%)? More generally, for one DIA run, draw the ROC curve (x-axis: FDR, y-axis: precursors#) for each tool, which will show the global comparison. These figures can be put in the supplementary materials.

2. Supplementary Figure 1a is library-based analysis; 1b is library-free analysis. This is not correct (both are library-based). The correct way is: supplementary Figure 1a is FP-MSF hybrid; 1b is FP-MSF.

3. In the main text lines 226-227, MaxDIA gave unreasonably low numbers of quantified peptides in these data, and thus was excluded from plotting (see Supplementary Figure 2a and 2c). Is this a bug for MaxDIA?

4. In Table S1, it is 90 min gradient for 2020-Yeast; in the main text lines 213-216, it is 115 min gradient for 2020-Yeast. They are not consistent.

5. Table S2: in row 4, it is FP-MSF, not FF-MSF.
6. Compared to DIA-NN, there is still some improving space for MSFragger-DIA, e.g. Figure 4b, Figure 5a.
7. How to explain FP-MSF hybrid is better than FP-MSF?
8. In the main text lines 363-365, for nonspecific searches (e.g., HLA peptidome data), MSFragger-DIA is significantly slower. What is the reason? Is it because the search space too big?

Reviewer #1:

In the work, the group of Prof. Nesvizhskii reports MSFragger-DIA for DIA proteomic data analysis. MSFragger-DIA starts with a direct search of MS/MS spectra against the entire protein sequence database to generate a list of peptide candidates. Then, MSFragger-DIA traces peaks, extracts ion chromatograms, and detects features of all candidate peptides for each spectrum determined. Finally, MSFragger-DIA generates output files compatible with PeptideProphet and Percolator for rescoring and FDR estimation. In general, it proposes a qualification and quantification pipeline for DIA data analysis, supporting DIA analysis without DDA, phosphoproteomics, single cell proteomics, etc. However, the current benchmarking doesn't demonstrate the superior performance of MSFragger-DIA compared to the state-of-the-art software solutions.

Response: Thank you very much for the summarization and comments. To better demonstrate the performance of our tools, we have added new experiments in Figure 2, 3, S2, and S3. We have also included a recently released tool, Spectronaut 17, to ensure our benchmarking remains up to date. Our updated experiments encompass comprehensive analyses from various perspectives, including sensitivity, false discovery rate control, overlap percentages among results from all tools, precision, and accuracy. Additionally, we have integrated another new tool, the latest version of EncyclopeDIA, into the benchmarking presented in Figures 3 and S3. Further evaluations of quantification precision have been conducted as well.

It is also not clear what is the main innovation point of the current work. Is it just to provide another software solution or are there key advances of the pipeline? The most concept of the current work, including fragment ion indexing, has been proposed by the group or others in the field.

Response: There are two aspects of the innovations. Firstly, we proposed a method and a new tool, MSFragger-DIA, to directly search the DIA spectra to identify the peptides. As outlined in the introduction section, current methods necessitate either spectral libraries or chromatography feature detections, both of which are time-consuming and costly. Our study demonstrates that it is feasible to obtain good results by directly searching the DIA spectra without relying on experimental or *in silico* spectral libraries. Secondly, we have

created a comprehensive software suite designed to streamline and ensure the full reproducibility of DIA data analysis. Our new FragPipe, in conjunction with MSFragger-DIA, has already facilitated the researcher to perform DIA data analysis easily. We have highlighted the innovations in the abstract (line 23-29), introduction (line 110-114), and discussion (line 466-474).

The specific comments:

1. There is a lack of comprehensive comparison with the current state-of-the-art software solutions. For comparison, it is necessary to show coverage, reproducibility, quantification accuracy and computation time. It is also necessary to show the consistency among results, i.e. how many proteins shared, how many gain, and how many lose? However, in the current work, the authors show only one or two aspects per dataset comparing to one or two other software solutions, which is not comprehensive enough.

Response: Thanks for pointing this out. We have added more comparisons and analyses to Figure 2 and 3. In the new analysis, there are upset plot (Figure 2a) to demonstrate the overlap among all results, box plot (Figure 2b) to evaluate the false positives and sensitivity, violin plot (Figure 2c) to compare the quantification precision among all tools, and LFQbench-style plot (Figure 2d) to benchmark the quantification accuracy. There is also additional supporting evidence in Figure S2 and Table S2. In Figure 3b, we have added the boxplot to show the quantification precision from the overlapping peptides and unique peptides. In the phosphoproteomics data analysis (Figure 5), we also use the upset plot, which is used to evaluate the overlap, to replace the bar plot. All those new analyses and figures demonstrate the excellent performance of our tools from the aspects of quantification sensitivity, precision, accuracy, and false discovery controls. We have also added the description to line 172-237, line 260-271, line 504-507, line 518-537, line 540-558, line 570-578.

2. For speed comparison, the MSFragger-DIA starts from mzml while the others start from raw, which is not a fair comparison.

Response: We apologize for any confusion caused. We used the same mzML files for DIA-Umpire, EncyclopeDIA, DIA-NN, and MSFragger-DIA for fair comparison. DIA-NN

does not support the raw format in Linux, while EncyclopeDIA lacks raw format support in all operating systems. We used the raw files for MaxDIA and Spectronaut as they exhibited better performance with raw files. The time required for format conversion was minimal, at approximately 7 minutes in total, compared to the runtime for the fastest tool, which is around 100 minutes (Figure 6). We have added the clarification to line 530-534, line 319-321, and line 351-353.

3. The authors name MSFragger-DIA as a one-stop analysis pipeline, but it needs DIA-NN for quantification, R packages for normalization and ProteoWizard for data conversion and demultiplexing, which is not one-stop at all.

Response: As emphasized in the title and main context, the one-stop analysis pipeline encompasses MSFragger-DIA and FragPipe. Users are not required to install DIA-NN, as the quant module is already integrated into FragPipe. The R script serves to filter the results and calculate the MaxLFQ intensities, ensuring fair comparisons. In real-world applications, users do not need to execute any R scripts while utilizing our tools. Furthermore, ProteoWizard is not necessary for un-staggered data. We have added clarifications to line 527-537. We have also changed the title by removing “one-stop”.

4. Comparison with spectronaut, the authors compared to different version of spectronaut, from version 13 to version 15. These versions are quite old. The newest version of spectronaut 16 with directDIA2.0 performs much better than these versions. The current comparison is miss leading.

Response: We appreciate your suggestion. We have incorporated the results from the most recent Spectronaut 17, which contains directDIA+, to Figure 2, Figure 3, and runtime comparison (line 356-357, line 504-507, and line 683-685). Unfortunately, we do not have access to Spectronaut for the other datasets, as it is a commercial software and is not available to us.

5. The current work doesn't show any benchmarking on timsTOF pro data, which is nowadays a main platform for proteomics research, and should be supported by the software.

Response: At present, DIA with timsTOF Pro (diaPASEF) represents a rapidly evolving area of research. Various research groups continue to develop new protocols tailored to different applications. For instance, in recent months, three research groups have proposed distinct acquisition protocols: Slice-PASEF, midiaPASEF, and Synchro-PASEF. Each protocol requires the development of a new algorithm for analysis, which is not a trivial task and requires big efforts. Therefore, incorporating the support for timsTOF data is beyond the scope of this manuscript. In this study, we have introduced a novel direct database searching method for traditional DIA data and a comprehensive DIA data analysis pipeline. We have included the discussions to line 457-462.

6. Comparison with MaxDIA: it mentions that on one dataset, MaxDIA shows low numbers of quantified peptides and on the other dataset, MaxDIA shows a very high rate of missing quantification values. Is it common on different datasets? or is it just specific to the two datasets? What can be the possible reason?

Response: We are uncertain about the underlying cause. We conducted the experiments once again using an updated version of MaxDIA (MaxQuant version 2.2.0.0); however, the results remained largely unchanged. As MaxDIA is a closed-source tool developed by a separate group, we lack the capability to investigate and debug the tool. For simplicity, we decided to remove the comparison with MaxDIA from that specific experiment (staggered-windows DIA with gas phase fractionation (GPF)).

7. Comparison with DIA-NN: we tested DIA-NN and MSFragger-DIA on the dataset ccRCC, which is used for generating Figure 6. Without missing value filtering, DIA-NN in silico library identified 9585 unique genes while FP-MSF identified 8317 unique genes in our result. Even after 50% missing value filtering, DIA-NN in silico library identified 7700 protein groups, 10% more than that shows in Figure 6. In our results, the missing values by DIA-NN and FP-MSF are comparable. Considering computation time, including that for data conversation, the computation time by FP-MSF is faster than DIA-NN in silico library but not significant when analyzing all the 187 files. Our result is not consistent with that in the paper.

Response: Due to the lack of detailed information regarding the parameters employed by the reviewer when running the tools, we are unable to provide extensive comments. As outlined in the "Software Availability" and "Data Availability" sections of our manuscript,

our parameters, results, and log files are publicly accessible at <https://doi.org/10.5281/zenodo.7261712>. Additionally, the scripts for counting numbers and generating figures can be found at <https://github.com/Nesvilab/MSFragger-DIA-manuscript>. We have modified Figure 6a to display the number of quantified proteins since MaxDIA does not provide gene-level reports. However, this alteration does not affect our conclusion. We have also added the bar plots without discarding the proteins with more than 50% missing values to Supplementary Figure S5. The reviewer is welcome to examine and reproduce our results.

8. FDR evaluation, it is suggested to test the FDR using the cohort data (ccRCC) and using entrapment sequences, e.g. *E. coli*, yeast, *A. thaliana*. The current benchmarking is not enough.

Response: Thank you very much for the suggestion. We have added the entrapment database searching with the 20 ccRCC dataset. To ensure a sufficient number of entrapment sequences, we combined databases of *H. sapiens*, *S. cerevisiae*, *E. coli*, and *A. thaliana*. Details can be found in line 359-361 and line 666-677.

At the meantime, Figure 2 and Table S2 already have similar analyses. The database was from the combination of human and *E. coli* proteome, but the “Lymphnode” condition only has human samples. Thus, the numbers of the *E. coli* precursors in the “Lymphnode” condition were used to evaluate the false discover rates.

It mentions that “Replacing the experimental fragment peaks with the predicted peaks resulted in a lower actual FDR (Supplementary Table S2).” What is the reason? Normally, experimental library is more specific to the DIA data and should perform better.

Response: Thank you for asking. To clarify any potential misunderstandings, we do not replace the entire experimental library with the *in silico* library. Instead, we only replace the fragment peaks in the experimental library with the predicted peaks, while retaining the original peptide sequences, modifications, and retention times. The rationale behind this approach is that, in the case of low-abundance peptides, the experimental fragment peaks frequently appear incomplete, with some peptides exhibiting only a few fragment peaks. Also, low-quality peptides may contain interfering peaks, leading to an increased likelihood of false matches. The incompleteness and interference make it challenging for

the target-decoy modeling to accurately distinguish true matches from false ones. By replacing the experimental fragment peaks with the *in silico* predicted peaks, we address this issue of incompleteness, thus improving the model's performance. We have added the explanation to line 189-200.

9. Single cell proteomics, the current protein identification number is lower than many state-of-the-art reports (<https://doi.org/10.1101/2022.06.28.498038>), which of course is related to the dataset. The authors should choose timsTOF pro datasets for test, where > 2000 or even 3000 proteins were reported for single cell proteomics.

Response: Thanks for the information. The dataset referred to by the reviewer is entirely distinct, originating from a different mass spectrometer and incorporating additional ion mobility separations. We believe that comparing this dataset with the ones used in our manuscript would not be a fair assessment. As a manuscript focusing on the computational method development, the comparisons should use the same dataset and settings.

10. Phosphoproteomics, synthetic phosphopeptides dataset (e.g. Nature Communications 2020, 11, 787) should be adopted to demonstrate the false localization rate of phosphate. For phosphoproteomics, it should be compared to spectronaut instead of DIA-NN, where the latter is not yet optimized for phosphoproteomics.

Response: In this manuscript, we proposed a direct database searching method, and a comprehensive data analysis suite for DIA data. Thus, Phosphorylation site localization is beyond the scope of this manuscript.

A recent paper (<https://doi.org/10.1038/s41467-022-35740-1>) revealed that Spectronaut exhibited high false ID percentages (Figure 4 of the publication). Our comparisons using a different dataset (Figure 2b and Table S2) also gave the similar conclusion. Therefore, we believe it is appropriate to include DIA-NN in the comparison.

11. Different methods were suggested for MSFragger-DIA, those with and without DDA data library, and with in-silico library. It should be compared the identification consistency and

quantification accuracy of the different methods, and it should be discussed which one is recommended under what condition.

Response: Thanks for the suggestions. In Figure 2, 4 and 6, we compared the DIA-only and hybrid approaches. We also added the discussions about different strategies in line 434-452.

Minor: gas phase fractionation (GFP) should be GPF.

Response: We apologize for the oversight and have made the corrections.

Reviewer #2:

The reviewer would like to answer the following questions:

What are the noteworthy results?

A fast and sensitive platform for DIA identification and quantification.

Will the work be of significance to the field and related fields? How does it compare to the established literature? If the work is not original, please provide relevant references.

Yes, the work is of significance. Comparing to in silico library-based DIA-NN, MSFragger-DIA is faster, and has similar or better DIA quantification results.

Does the work support the conclusions and claims, or is additional evidence needed?

Yes, it does.

Are there any flaws in the data analysis, interpretation and conclusions? Do these prohibit publication or require revision?

Yes, please see the below comments for minor revision.

Is the methodology sound? Does the work meet the expected standards in your field?

Yes, it sounds good, and meet the expected standards.

Is there enough detail provided in the methods for the work to be reproduced?

Yes, it is.

The authors described the algorithm of MSFragger-DIA, compared to other tools (DIA-Umpire, Spectronaut, DIA-NN, MaxDIA), demonstrated the advantage of MSFragger-DIA (faster, similar or better than DIA-NN). The way of building libraries and the pipeline for identification make MSFragger-DIA different from other tools. If the authors can follow the minor suggestions, the manuscript will become better.

Response: We sincerely appreciate your insightful evaluation of our manuscript and the recognition of MSFragger-DIA's advantages. We are grateful for your suggestions and comments, which will enhance the quality of our work. We have addressed all of the comments listed below.

Here are the minor suggestions.

1. Figure 2 shows FP-MSF and FP-MSF hybrid outperform DIA-NN and Spectronaut at 1% FDR. How about different FDR cutoff (e.g. 0.5%, 5%)? More generally, for one DIA run, draw the ROC curve (x-axis: FDR, y-axis: precursors#) for each tool, which will show the global comparison. These figures can be put in the supplementary materials.

Response: We appreciate your suggestions. Unfortunately, applying a series of FDR thresholds to filter the results is challenging. By default, FP-MSF and FP-MSF hybrid filter the spectral library at a 1% FDR, followed by another round of FDR filtering through the DIA-NN quantification module. The library-free DIA-NN approach also involves multiple rounds of FDR filtering during peptide identification and quantification, with some defaulting to a 1% FDR. Spectronaut enforces a 1% FDR threshold by default, and altering it necessitates re-running the entire analysis. Given these constraints, fairly filtering all results with FDRs other than 1% is highly difficult.

2. Supplementary Figure 1a is library-based analysis; 1b is library-free analysis. This is not correct (both are library-based). The correct way is: supplementary Figure 1a is FP-MSF hybrid; 1b is FP-MSF.

Response: Sorry for the confusion. Supplementary Figure 1a is library-based because it requires DDA data as a library. Supplementary Figure 1b is library-free because it does not require DDA data or pre-generated library. Supplementary Figure 1a is not hybrid

because the DIA data is not used in the internal library building step. We have added calcifications to line 434-443, and the Supplementary Figure 1's figure legend.

3. In the main text lines 226-227, MaxDIA gave unreasonably low numbers of quantified peptides in these data, and thus was excluded from plotting (see Supplementary Figure 2a and 2c). Is this a bug for MaxDIA?

Response: We have no clue about the reason because MaxDIA is a closed source tool from another group. We also re-ran the experiment using a newer version (2.2.0.0) but still got similar results. For simplicity, we decided to remove the comparison with MaxDIA from that specific experiment (staggered-windows DIA with gas phase fractionation (GPF)).

4. In Table S1, it is 90 min gradient for 2020-Yeast; in the main text lines 213-216, it is 115 min gradient for 2020-Yeast. They are not consistent.

Response: Thank you for pointing this out. We have corrected them.

5. Table S2: in row 4, it is FP-MSF, not FF-MSF.

Response: Thank you for pointing this out. We have corrected it.

6. Compared to DIA-NN, there is still some improving space for MSFragger-DIA, e.g. Figure 4b, Figure 5a.

Response: Yes, we agree that there is still room for improvement. We will continue to enhance and upgrade MSFragger-DIA and FragPipe, as we have been committed to doing over the past years.

7. How to explain FP-MSF hybrid is better than FP-MSF?

Response: The library from the FP-MSF hybrid approach contains a greater number of high-quality spectra due to the incorporation of DIA data in the library build, which is not present in the FP-MSF approach. We have expanded our discussions on this topic in line 440-452.

8. In the main text lines 363-365, for nonspecific searches (e.g., HLA peptidome data), MSFragger-DIA is significantly slower. What is the reason? Is it because the search space too big?

Response: Yes, it is because the search space is very big. MSFragger-DIA needs to search all peptides within a much wider mass window compared to DDA database search.

REVIEWERS' COMMENTS

Reviewer #1 (Remarks to the Author):

In this revision, the authors have significantly improved the manuscript by adding additional benchmarking analysis, which is appreciated. Most of my previous concerns have been addressed. However, I still want to see more analysis on the phosphoproteomics part. As the authors used MSFragger-DIA for phosphoproteomics, it is necessary to demonstrate the accuracy in phosphate localization. The authors are recommended to analyze the benchmarking data by Dorte B. Bekker-Jensen (Nature Communications 2020, 11, 787) of synthetic phosphopeptides. The number of ID is only comparable when the false localization rates are controlled at the same level.

Reviewer #2 (Remarks to the Author):

The authors made significant changes to improve the quality. The latest version of Spectronaut was added to the comparison. The latest development of timsTOF was discussed. More details about the comparisons were explained. It is good for publication after resolving the following minor issues.

1. Figure 2a, which data are used (which condition: "Lymphnode", "1-25", "1-12", "1-06", or all four; which species: E. coli, H. sapiens, or both).
2. Supplementary Figure S3 figure legend, (a) The bar height is the average number of four replicates (not three).

Reviewer #1:

In this revision, the authors have significantly improved the manuscript by adding additional benchmarking analysis, which is appreciated. Most of my previous concerns have been addressed. However, I still want to see more analysis on the phosphoproteomics part. As the authors used MSFragger-DIA for phosphoproteomics, it is necessary to demonstrate the accuracy in phosphate localization. The authors are recommended to analyze the benchmarking data by Dorte B. Bekker-Jensen (Nature Communications 2020, 11, 787) of synthetic phosphopeptides. The number of ID is only comparable when the false localization rates are controlled at the same level.

Response: We greatly appreciate your review and feedback. It's gratifying to hear that our additional benchmarking analysis has been valuable.

We had indeed considered incorporating the synthetic phosphopeptides dataset from Dorte B. Bekker-Jensen's publication, as the reviewer suggested. Regrettably, after searching in the supplementary information and the ProteomeXchange repository PXD014525, neither the database file nor the synthetic peptide sequences could be found. Thus, we could not reproduce the experiment of that paper. Further, as our study primarily concentrates on peptide identification, the comprehensive investigation of modification localization is considered beyond its scope. However, we acknowledge the importance of the reviewer's point and have discussed this in a new paragraph within the manuscript. It reads: "*Since this study focuses on the peptide identification, we did not thoroughly investigate other aspects of the DIA data analysis such as modification localization. However, by integrating with FragPipe and DIA-NN, we can perform site localization in the DIA-NN quantification step [5]. Lou et al [70] demonstrated that the DIA-NN localization algorithm was more conservative than Spectronaut's but resulted in lower sensitivity. Considering the modification localization for the DIA data is still an open question, we will investigate it in the future work.*"

Reviewer #2:

The authors made significant changes to improve the quality. The latest version of Spectronaut was added to the comparison. The latest development of timsTOF was discussed. More details

about the comparisons were explained. It is good for publication after resolving the following minor issues.

Response: Thank you very much for the summarization and comments. We are glad to hear that this manuscript is good for publication. Please find our point-to-point responses in the following.

1. Figure 2a, which data are used (which condition: “Lymphnode”, “1-25”, “1-12”, “1-06”, or all four; which species: *E. coli*, *H. sapiens*, or both).

Response: Sorry for the confusion. It is from all four conditions of both species. We have added the clarification to the figure legend: “*The precursors are from all four conditions of both H. sapiens and E. coli.*”.

2. Supplementary Figure S3 figure legend, (a) The bar height is the average number of four replicates (not three).

Response: Sorry for the oversight. We have corrected the mistake.